# Hyperhomocysteinemia in Patients with Newly Diagnosed Primary Hypertension in Can Tho City, Vietnam

**DOI:** 10.3390/healthcare11020234

**Published:** 2023-01-12

**Authors:** Son Kim Tran, Toan Hoang Ngo, Phi Hoang Nguyen, An Bao Truong, Giang Khanh Truong, Khoa Dang Dang Tran, Phuong Minh Vo, Phi The Nguyen, Thuan Tuan Nguyen, Phu Ngoc Thien Nguyen, Kien Trung Nguyen, Hung Do Tran

**Affiliations:** 1Department of Internal Medicine, Can Tho University of Medicine and Pharmacy, Can Tho 900000, Vietnam; 2Department of Cardiology, An Giang Cardiovascular Hospital, Long Xuyen 880000, Vietnam; 3Faculty of Medicine, Can Tho University of Medicine and Pharmacy, Can Tho 900000, Vietnam; 4Faculty of Nursing and Medical Technology, Can Tho University of Medicine and Pharmacy, Can Tho 900000, Vietnam

**Keywords:** homocysteine, hypertension, newly, diagnosed, primary

## Abstract

Background: Elevated levels of blood total homocysteine is one of the cardiovascular risk factors in hypertensive patients. Objectives: Determine the prevalence of hyperhomocysteinemia and its associated factors in newly diagnosed primary hypertension patients. Materials and methods: A cross-sectional descriptive study on 105 patients with newly diagnosed primary hypertension at Can Tho University of Medicine and Pharmacy Hospital from May 2017 to May 2018. Total homocysteine levels and related factors were collected at the study time. Results: The mean plasma total homocysteine level was 16.24 ± 4.49 µmol/L. There were 78 patients with elevated plasma total homocysteine levels ≥15 µmol/L, accounting for 74.3% of all patients. Being elderly, gender, hypertension stage, and diabetes were factors associated with hyperhomocysteinemia (*p* < 0.05). Total homocysteine levels were positively correlated with SBP, DBP, and age with r(SBP) = 0.696, r(DBP) = 0.585, and r(age) = 0.286. Conclusion: Research on the subpopulation of Vietnamese people shows that hyperhomocysteinemia is common in patients with newly diagnosed primary hypertension, and high blood total homocysteine levels are often related to age, sex, hypertension stage, and diabetes.

## 1. Introduction

In addition to being the leading cause of cardiovascular disease in many countries, hypertension is a common medical issue encountered in the clinic. In 2017, it was estimated that 34% of adults in the United States over the age of 20, or 85.7 million people, had hypertension [1]. The prevalence of hypertension in Viet Nam was 25.1%, or 11 million persons, according to an epidemiological survey in 2015 [2]. The prevalence of hypertension is rising continuously. Currently, there are approximately 1 billion people with hypertension around the world. It is anticipated that, by 2025, there will be 1.5 billion individuals with hypertension [3].

Many domestic and international authors have recently focused on homocysteine as an independent risk factor for cardiovascular disease [4,5]. Homocysteine is a sulfur-containing amino acid created during methionine metabolism and excreted in the urine [6]. Numerous clinical and epidemiological studies conducted by international researchers have demonstrated a correlation between plasma homocysteine levels and blood pressure, especially systolic blood pressure [7,8]. As a result, a homocysteine blood test can aid cardiologists in the early diagnosis and prognosis of hypertension. Therefore, we conducted this study with the following objective: to determine the prevalence and the factors associated with hyperhomocysteinemia in patients with newly diagnosed primary hypertension at the Can Tho University of Medicine and Pharmacy Hospital in 2017–2018.

## 2. Materials and Methods

### 2.1. Study Population

#### 2.1.1. Materials

All newly diagnosed primary hypertension patients were examined at the Can Tho University of Medicine and Pharmacy Hospital from May 2017 to May 2018.

#### 2.1.2. Inclusion Criteria

All patients were newly diagnosed with primary hypertension according to the JNC 6 criteria [9], similar to the ESC diagnosis and classification criteria [10].

#### 2.1.3. Exclusion Criteria

All patients with primary hypertension who had comorbidities that affected their plasma total homocysteine levels, including a history of liver disease, kidney disease, stroke, and chronic comorbidities (gout, rheumatoid arthritis, Parkinson’s disease), individuals are being treated with vitamin B6, B12, folate, and diabetic patients, are taking sulfonylurea (Figure 1) [11,12].

### 2.2. Methods

#### 2.2.1. Study Design

A cross-sectional descriptive study.

#### 2.2.2. Sample Size

A convenient sampling of all patients treated at Can Tho University of Medicine and Pharmacy Hospital who met the selection criteria during the study period was selected. In actuality, we conducted the study with 105 subjects.

### 2.3. Data Collection

The mean plasma total homocysteine level is in units of μmol/L, and hyperhomocysteinemia was identified when the plasma total homocysteine level is ≥15 mol/L [13]. Several characteristics, including gender, age, smoking, alcohol consumption, exercise habits, being overweight/obesity, diabetes, dyslipidemia, and echocardiographic left ventricular hypertrophy were associated with hyperhomocysteinemia.

#### 2.3.1. Data Analysis

The mean fasting total Hcy concentration (X + SD), which was typically within the range of 5 -<15 μmol/L, increases if >15 μmol/L [13]. The age was calculated by subtracting the year of birth from the survey year, and then participants were divided into two groups: ≥60 and <60 years old. Gender was divided into two groups, male and female. Being overweight/obesity was determined if BMI ≥23 kg/m^2^ according to BMI applied to Asians by WHO in 2000 [14]. 

Diabetes was diagnosed based on the ADA 2017 criteria (similar to the ADA 2021 criteria) if one of the following conditions was met: HbA1c ≥6.5% or fasting blood glucose level ≥7.0 mmol/L (126 mg/dL) or random plasma glucose ≥200 mg/dL (11.1 mmol/L) in patients with classic symptoms of hyperglycemia (polyphagia, polydipsia, polyuria, weight loss) or if the patient has been diagnosed with diabetes [15,16]. 

Hypertension was classified into three grades according to the JNC 6 criteria [13], similar to the 2021 ESC criteria for hypertensive classification [14]. According to NCEP-ATP III (2001) criteria, dyslipidemia was diagnosed if there was an abnormality in one of the following blood lipid components: total cholesterol > 5.2 mmol/L, blood triglycerides > 1.7 mmol/L, HDL-c <1.03 mmol/L in men or <1.29 mmol/L in women, LDL-c >3.4 mmol/L [17]. The American Society of Ultrasound defined ventricular hypertrophy as the left ventricular mass index (LVMI) >134 g/m^2^ for men or >110 g/m^2^ for women [18].

#### 2.3.2. Measurements

Blood pressure was measured using a Japanese ALPK2 watch and stethoscope. During the measurement, the patient rested for 15 min and refrained from using stimulants and talking. SBP corresponds to the first pulse sound (phase I Korotkoff), and DBP corresponds to its disappearance (phase V Korotkoff). Three measurements of blood pressure were taken 2 to 5 min apart, and the average of the three readings was obtained [19].

Height (m) and weight (kg) were measured with a tape measure and a standardized scale. Using the formula weight (kg)/height, the BMI was computed (m^2^) [20]. 

On echocardiography, left ventricular hypertrophy was measured with a Siemens X500 ultrasound device. Assessment of left ventricular mass (LVM) was performed using Devereux’s formula, LVM (g) = 1.04 ((LVDd + IVSd + PWLVd)³ − LVDd³) − 13.6 g. Based on the Mosteller formula, the left ventricular mass index (LVMI) was calculated by adjusting it according to the body surface area (BSA): LVMI (g/m^2^) = LVM (g)/BSA (m^2^), with BSA (m^2^) = √((height (cm) × weight (kg))/3600) [18].

Using polarized fluorescence and the principle of competitive immunoassay, the Abbott Diagnostics Axsym instrument quantified the plasma THcy concentration based on the theory of competitive immunoassay [21]. Glucose, total cholesterol, triglycerides, HDL-c, and LDL-c concentrations were determined using Cobas-e automatic biochemical analysis by enzymatic colorimetric technique [22,23]. Quantification of HbA1C was performed via the plasma turbidity immunoassay technique utilizing the ARCHITECT i2000R [24].

#### 2.3.3. Statistical Analysis

Computerized data processing was conducted using SPSS 20.0 software. The format for presenting quantitative variables is mean standard deviation. Frequencies and percentages (%) represent qualitative variables (%). To compare the difference between qualitative variables, we utilized the chi-squared test and adjusted it according to Fisher’s exact test for tables with more than 25 percent expected value <5. Enter technique multivariate logistic regression analysis to uncover factors strongly connected with homocysteine level. For quantitative variables having a normal distribution, use Pearson’s correlation coefficient. A *p*-value of 0.05 or lower is generally considered statistically significant.

## 3. Results

### 3.1. Baseline Subject Characteristics

The study on 105 patients included 29 men (27.6%) and 76 women (72.4%); the mean age was 63.07 ± 9.32; the mean BMI was 23.46 ± 3.23 Kg/m^2^, and the mean tHcy concentration was 16.24 ± 4.49 μmol/L. The rate of hypertension in grade 1 was 61%, in grade 2 was 17.1%, and in grade 3 was 21.9%. The proportion of people aged 60 and over was 68.6%, the percentage of people who are overweight was 57.1%, the proportion of people with diabetes was 46.7%, the rate of people with dyslipidemia was 89.5%, and the rate of people with left ventricular hypertrophy on echocardiogram was 23.8% (Table 1).

### 3.2. Prevalence of Hyperhomocysteinemia in Patients with Primary Hypertension

The prevalence of hyperhomocysteinemia in our study was 74.3%, and men were taller than women; 60 and above accounted for the highest percentage by age group (Figure 2).

### 3.3. Some Factors Associated with Hyperhomocysteinemia in Patients with Primary Hypertension

There were a statistically significant relationship between hyperhomocysteinemia and gender, age group, diabetes, hypertension stage, and metformin therapy (*p* < 0.05) (Table 2). Multivariable logistic regression analysis demonstrates that diabetes was indeed associated with the rate of hyperhomocysteinemia (*p* < 0.05) (Table 3). The correlation between the concentration of tHcy and age was weak, with a r-value of 0.28 (*p* < 0.05). The concentration of tHcy had a strong positive correlation with SBP and DBP, with r values of 0.69 and 0.58, respectively (*p* < 0.001) (Table 4). Univariate linear regression equation between plasma homocysteine concentration and age (R2 = 0.082): Total homocysteine = 6.1977 + 0.1852 * Age (Figure 3) and univariate linear regression equation between plasma total homocysteine concentration and SBP (R2 = 0.4846): SBP = 123.33 + 2.206 * Total homocysteine (Figure 4).

## 4. Discussion

### 4.1. Prevalence of Hyperhomocysteinemia in Patients with Primary Hypertension

Our study was conducted on 105 primary hypertensive patients being treated at the Can Tho University of Medicine and Pharmacy Hospital, with the mean age of study subjects was being 63.07 ± 9.32 years old. The highest concentration of study subjects was in the age group ≥60 years old (67.6%). This is the age when people are most likely to develop cardiovascular disease in general, and hypertension in particular [25].

The relationship between tHcy and hypertension has been proven experimentally. Many studies have shown that tHcy is an independent risk factor for cardiovascular disease and can act as a promoter of hypertension through mechanisms such as smooth muscle hypertrophy, decreased cellular function of smooth muscle, damage to vascular endothelial cells, and changes in vasomotor regulation leading to hardening of the vessel wall [26]. Sutton-Tyrrell K. et al. (1997) conducted a study to demonstrate that high homocysteine levels were an independent risk factor for hypertension in adults. The results showed that the mean age in the hypertensive group was 75.3 years old, the homocysteine concentration in the hypertensive group was 11.5 mol/L, and the non-hypertensive group was 9.9 mol/L with *p* < 0.001. The author concluded that plasma homocysteine concentration is associated with SBP with an OR = 2.1 (95% CI; 1.1–3.8) [27]. In addition, our study also showed that the proportion of patients with increased plasma tHcy also tended to increase gradually according to the degree of hypertension.

### 4.2. Some Factors Associated with Hyperhomocysteinemia in Patients with Newly Diagnosed Primary Hypertension

We discovered that the prevalence of hyperhomocysteinemia was higher in the male participants and in the age group ≥60 years old (*p* < 0.05). An epidemiological study by Framingham (2004) was carried out on 1160 subjects in the community after measuring the concentration of complete plasma tHcy, and Framingham came to the conclusion that plasma tHcy concentration in males was higher than in females. In addition, this concentration value also rises with age. The age-specific increase was statistically significant (*p* < 0.001) for both men and women right after adjusting for blood vitamin levels [28]. NHANES III data also indicate that serum tHcy levels grow with age and that there was little variation between ethnic groups [29]. This correlation may be attributable to genetic, heritable, and environmental variables in the tHcy metabolic pathway. Numerous demographic studies undertaken in Europe, America, Africa, and Asia all reach the same conclusion: plasma tHcy levels were higher in males than in females, and they rise with age [30,31,32,33].

Being overweight/obesity is a risk factor for high blood pressure status, as shown by the prevalence of hypertension in obese people being double that of non-obese individuals [34]. According to pathophysiology, obese people are frequently accompanied by severe fatty liver disease, which impairs liver function and alters enzyme activity in the tHcy metabolic pathway. Consequently, this influences the changes in tHcy concentration. Multiple studies have demonstrated a positive correlation between BMI and plasma tHcy levels. The observation by Karatela (2009) showed that, among overweight and obese subjects, plasma tHcy levels were significantly increased, along with decreased concentrations of vitamin B12 and folic acid compared with normal weight subjects [35]. In addition, blood pressure levels were also higher in obese subjects compared with normal people. Therefore, when BMI increases, it is also associated with an increase in plasma tHcy. However, due to the small sample size, our study did not record the relationship between overweight and obesity and plasma homocysteine levels. Further studies with larger sample sizes are needed to clarify this connection.

Nearly 90% of subjects in our sample population had dyslipidemia. The study conducted by Ningjun Li et al. suggested that tHcy induces atherosclerosis by stimulating the proliferation of vascular smooth muscle cells. In addition, tHcy also had the effect of increasing cholesterol oxidation, which can cause high blood vessel atherosclerosis, promoting lipid peroxidation and LDL oxidation in the body. These factors may contribute to the process of atherosclerosis [36]. Perhaps due to the small sample size, through our study, we had not found a relationship between tHcy concentration and dyslipidemia. However, in the world, there have been many studies showing the relationship between dyslipidemia and increased tHcy [37,38].

Left ventricular hypertrophy will gradually increase over time and depend on the degree of hypertension, while the plasma tHcy level will only increase gradually with the degree of hypertension. According to the study of Yawen Deng et al. (2022), when retrospectively analyzing the association between hyperhomocysteinemia and left ventricular hypertrophy in hypertensive patients, the authors found that an increased rate of homocysteine accounted for a higher proportion in the group of hypertensive patients with left ventricular hypertrophy. Furthermore, hyperhomocysteinemia was significantly associated with the presence of left ventricular hypertrophy in hypertensive patients [39]. However, the correlation between the left ventricular hypertrophy index (LVMI) on echocardiography and the rise in plasma tHcy levels was not statistically significant (*p* = 0.072) according to our study. Perhaps because our study subjects were newly diagnosed hypertensive patients, most of them were grade 1 hypertension (60.9%) and did not have enough time to progress to left ventricular hypertrophy.

Even so, our study results show that there was still a statistically significant association between diabetes mellitus and increased plasma tHcy levels (*p* = 0.049). Pathophysiologically, the direct biochemical consequences of hyperglycemia included: altered endothelial function, indirect influence on endothelial cell activity through the synthesis of growth factors, and cytokines and vasoactive functions in other cells are prerequisites for atherosclerotic plaque formation. Patients with diabetes are often prone to cardiovascular diseases and are often accompanied by increased plasma tHcy, which are factors that can participate in the process of atherosclerosis [40].

Investigating the correlation of homocysteine with some clinical and biochemical characteristics showed that plasma tHcy concentration was correlated with SBP and age, with correlation coefficient r being, respectively: r(age) = 0.286 and r(SBP)) = 0.696 (*p* < 0.05). Thereby, we built the regression equation of homocysteine by age: total homocysteine = 6.1977 + 0.1852 * age. From this equation, we found that, for every 10 years of increase, homocystine concentration increased by 1.8 μmol/L. In the research of Henry O.R. et al. (2012) and in Jackson’s cardiovascular study on the correlation between homocysteine levels with vitamin B12 and folic acid, the authors conducted a study on 5129 African Americans, aged 21–94 years. The mean was 55 ± 13 years old, the proportion of male was 37% and female was 63%. The authors concluded that, for every 10 years increase, the plasma homocysteine concentration increased by 0.8 µmol/L with *p* < 0.001 [41]. This result was completely consistent with our study. Besides, we also built the regression equation of SBP according to homocysteine concentration: SBP = 123.33 + 2.206 * total homocysteine. According to this equation, for every 5 μmol/L increase in plasma homocysteine concentration, SBP increases by about 10 mmHg. Similarly, this result was perfectly congruent with the findings of Lim U (2002) and Wu (2018) [42,43]. The mechanism that is explained for the correlation between the plasma tHcy and SBP was that when plasma tHcy concentration increases, it causes narrowing of small arteries, renal dysfunction, increased sodium absorption, and arteriosclerosis [44]. THcy is an amino acid that contains sulfur as one of the main ingredients. Because the kidney is involved in the metabolism of a significant amount of tHcy, it is an important organ in the regulation of this amino acid when its plasma concentration is elevated. High plasma tHcy is a predictor of the risk of kidney damage, hypertension, and cardiovascular disease. There are a number of harmful effects associated with increased plasma tHcy levels, one of which is increased oxidative stress that damages endothelial cells [45]. Numerous clinical studies conducted worldwide over the past decade have demonstrated a positive correlation between THcy concentrations and blood pressure levels [46,47].

#### Limitations

This study also has certain limitations. It must be noted that all samples were collected from the same hospital, and the sample size was rather small, thus the results cannot be generalized to the majority of patients in Vietnam. Since the study’s sample size was limited and the individuals were recently diagnosed with hypertension, the majority (60.9%) had grade 1 hypertension and further testing for vitamin B12 and folate and analysis of the relationship between these factors and hyperhomocysteinemia are required. As a result, a number of cardiovascular risk factors have not been linked to increased homocysteine levels. Larger and multicenter investigations are required to overcome our study’s shortcomings.

## 5. Conclusion

Research in the subgroup of the Vietnamese population shows that the prevalence of hyperhomocysteinemia is relatively high in people with primary hypertension. Hyperhomocysteinemia is influenced by gender, age, hypertension stage, and diabetes.

## Figures and Tables

**Figure 1 healthcare-11-00234-f001:**
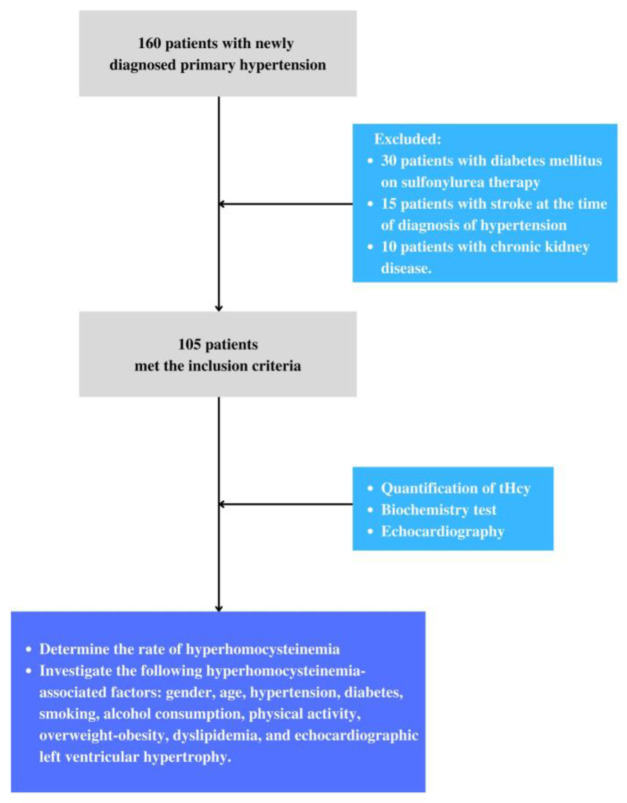
Participant flow diagram.

**Figure 2 healthcare-11-00234-f002:**
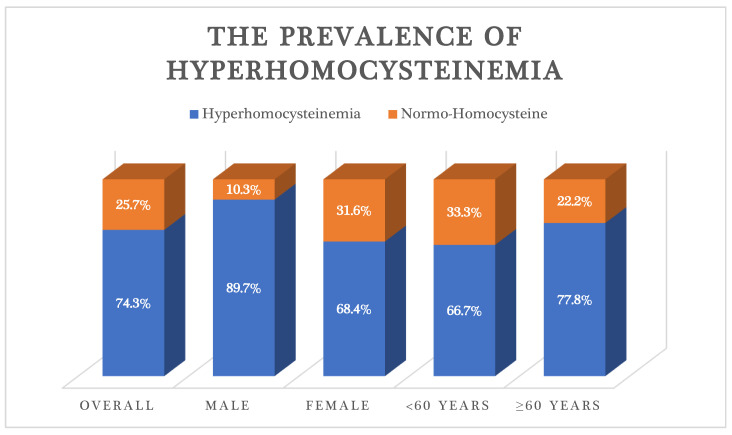
The prevalence of hyperhomocysteinemia in patients with primary hypertension be gender and age group.

**Figure 3 healthcare-11-00234-f003:**
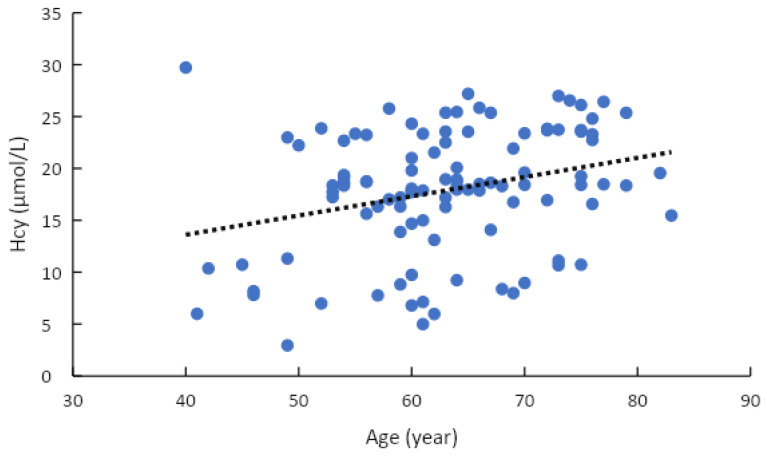
Plasma total homocysteine concentration and age correlation.

**Figure 4 healthcare-11-00234-f004:**
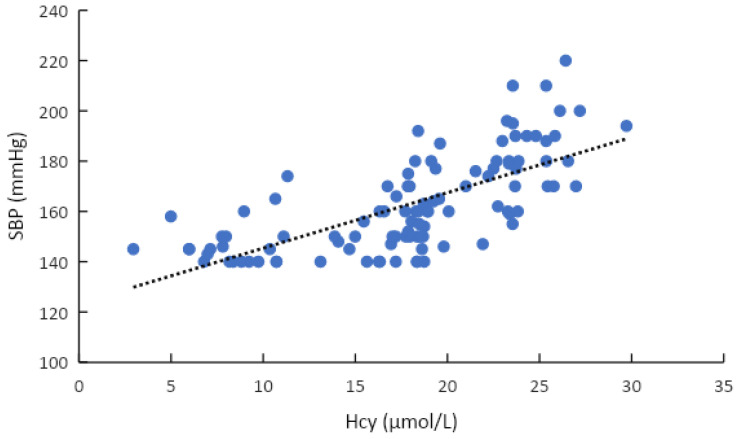
Plasma total homocysteine concentration and systolic blood pressure correlation.

**Table 1 healthcare-11-00234-t001:** Baseline characteristics of the study population.

Characteristics	Mean ± SD or n (%)
Age (years)	63.07 ± 9.32
BMI	23.46 ± 3.23
Total homocysteine (µmol/L)	17.88 ± 6.03
Male	29 (27.6)
Grade 1 hypertension	64 (60.9)
Grade 2 hypertension	18 (17.1)
Grade 3 hypertension	23 (21.9)
Age ≥ 60	72 (68.6)
Diabetes	49 (46.7)
Overweight-obesity	60 (57.1)
Dyslipidemia	94 (89.5)
Left ventricular hypertrophy on echocardiogram	25 (23.8)
Antiplateles	15 (14.29)
Statin	94 (89.5)
Metformin	33 (31.43)

**Table 2 healthcare-11-00234-t002:** The relationship between some risk factors and hyperhomocysteinemia.

Factor		Hyperhomocysteinemia		OR	95% CI	*p **
		Yes	No			
Gender	Male	26 (89.7)	3 (10.3)	4.00	1.10–14.52	0.026
	Female	52 (68.4)	24 (31.6)			
Age ≥ 60	Yes	56 (77.8)	16 (22.2)	2.87	1.15–7.19	0.021
	No	22 (66.7)	11 (33.3)			
Diabetes	Yes	32 (65.3)	17 (34.7)	0.41	0.17–1.01	0.049
	No	46 (82.1)	10 (7.9)			
Overweight-obesity	Yes	46 (76.7)	14 (23.3)	1.34	0.55–3.22	0.519
	No	32 (71.1)	13 (28.9)			
Dyslipidemia	Yes	69 (73.4)	25 (25.6)	0.61	0.12–3.04	0.725
	No	9 (81.8)	2 (18.2)			
Hypertension	Grade 1	37 (57.8)	27 (42.2)	-	-	<0.001
	Grade 2	18 (100)	0 (0.0)			
	Grade 3	23 (100)	0 (0.0)			
Left ventricular hypertrophy on echocardiogram	Yes	22 (88.0)	3 (12.0)	3.14	0.86–11.49	0.072
	No	56 (70.0)	24 (30.0)			
Metformin therapy	Yes	20 (60.6)	13 (39.4)	2.693	1.08–6.69	0.03
	No	58 (80.6)	14 (19.4)			

* Chi-square test.

**Table 3 healthcare-11-00234-t003:** Multivariable logistic regression analysis between plasma total homocysteine and some related factors.

Factor		OR	95% CI	*p*
Gender	Male	5.143	0.894–29.575	0.067
	Female			
Elder	Yes	12.455	0.793–195.669	0.073
	No			
Diabetes	Yes	0.25	0.079–0.796	0.019
	No			
Hypertension	Grade 1	-	-	-
	Grade 2	-	-	0.998
	Grade 3	-	-	0.998

**Table 4 healthcare-11-00234-t004:** Correlation between plasma total homocysteine concentration with some clinical and biochemical features.

Factor	r	*p*
SBP *	0.696	<0.001
DBP *	0.585	<0.001
Age *	0.286	0.003
Fasting blood glucose *	−0.046	0.642
HbA1c *	−0.095	0.335
Metformin therapy **	−0.086	0.381

* The Pearson correlation, ** the Spearman’s correlation.

## Data Availability

The datasets generated and/or analyzed during the current study are available from the corresponding author on reasonable request.

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
