# Peer review of "Hyperhomocysteinemia in Patients with Newly Diagnosed Primary Hypertension in Can Tho City, Vietnam"

_healthcare, 2023, doi:10.3390/healthcare11020234_

Round 1

Reviewer 1 Report

The authors collected plasma Hcy data from 105 patients with newly diagnosed primary hypertension. The relationships with plasma Hcy levels and age, gender, hypertension, and DM have been independently reported in many other studies to date, but this is new data from in the subpopulation of Vietnamese people. Thus, this study has some degree of values in the standpoint of preventive medicine/healthcare.

1.      In the first sentence of the Abstract; an independent risk factor for what? Hypertension?

2.      In the last sentence of the Abstract; Conclusion should be “Homocysteinemia was relatively prevalent among individuals with primary hypertension and elevated tHcy levels were related to age, gender, hypertension and diabetes in the subpopulation of Vietnamese people.” Only 105 patients do not represent Vietnamese people.

3.      Most “homocysteine” should be “total homocysteine (tHcy)”.

4.      What is the significant digit? For example, the odd rate (Spell out OR somewhere!) for gender should be 4.00 but not 4.

5.      Do not divide Table 3 to two pages!

6.      Where does “4.2 Results of controlling serum…… fraction” come from? (lines 215-216).

7.      Discussion is too long compared to its contents and can be shortened.

Author Response

First, the authors sincerely thank the reviewers for agreeing to read and give us valuable contributions.

Here we would like to reply to the comments of the reviewer as follows:

  1. In the first sentence of the Abstract; an independent risk factor for what? Hypertension?

Response 1: We want to rewrite it for clarity: Homocysteine ​​is one of the cardiovascular risk factors in hypertensive patients.

  1. In the last sentence of the Abstract; Conclusion should be “Homocysteinemia was relatively prevalent among individuals with primary hypertension and elevated tHcy levels were related to age, gender, hypertension and diabetes in the subpopulation of Vietnamese people.” Only 105 patients do not represent Vietnamese people.

Response 2: We want to clarify: Research shows that hyperhomocysteinemia is common in patients with newly diagnosed primary hypertension, and high blood homocysteine ​​levels are often related to age, sex, hypertension stage, and diabetes. We want to add the location to the study name in Can Tho City, Vietnam.

  1. Most “homocysteine” should be “total homocysteine (tHcy)”.

Response 3: We agree with the mentions of the reviewer.

  1. What is the significant digit? For example, the odd rate (Spell out OR somewhere!) for gender should be 4.00 but not 4.

Response 4: We apologize; when we used the chi-square test, men and women, age over 60 years, diabetes, and stages of hypertension were statistically significant differences between the groups, with and without hyperhomocysteinemia. We will fix the paraphrase in table 2.

  1. Do not divide Table 3 to two pages!

Response 5: We will revise it again.

  1. Where does “4.2 Results of controlling serum…… fraction” come from? (lines 215-216).

Response 6: Sincerely sorry; we made an error when we copied the structure of MDPI from another article of ours that is currently being peer-reviewed at MDPI. We want to correct it as "Some factors associated with hyperhomocysteinemia in patients with newly diagnosed primary hypertension."

  1. Discussion is too long compared to its contents and can be shortened.

Response 7: We will trim the discussion according to the feedback from the reviewer

Finally, on behalf of the research team, we would like to thank the reviewer for reading carefully and giving us important contributions to make our article more complete.

Reviewer 2 Report

Even if the topic is always current due to the mortality by cardiovascular diseases that shows no signs of decreasing, it is not clear what is the novelty of that study.

The association between homocysteine and age and blood pressure are widely discussed in the literature. A cross-sectional study with a limited number of participants does not have such a value as to reconfirm the numerous studies already carried out, which the authors widely discuss.

More specifically, here are some tips:

- The purpose and innovation brought by this study should be better explained

- A power analysis is missing

- It is not clear how many enrolled individuals were excluded by the criteria

- The study of the characteristics of the sample is not sufficient, for example, there is no information about the medications taken by the participants. This is a critical point, considering the association with diabetes and taking into account that the association between the use of metformin and elevated homocysteine levels has been widely demonstrated in the literature

- There is no complete blood chemistry data. The levels of B vitamins, especially vitamin B12 and folate, could be decisive in explaining homocysteine levels and it is not easy to make any conclusions without such data. Mean corpuscular volume and haemoglobin levels could also help identify possible confounders

- There is no statistical evaluation that provides a model that identifies associations studied by covariates and adjusted for confounders (many of which I do not believe are in the authors' possession, as discussed above)

- It is not possible to evaluate an association with hypertension, as all the participants were admitted for this characteristic. The presence of healthy controls would have been more helpful for this purpose. The only association that the authors can make in this regard is on the severity of hypertension, even if the population sample is probably underpowered for this purpose

- Figure 2 is poor. It might be better to include histograms describing the frequency of hyperhomocysteinemia in various subgroups (by age, gender, etc.)

- The discussion is too verbose and focused towards mechanistic concepts which are not part of the scope of the work. Lines 215-216 are out of context.

- Some passages should be revised from a linguistic point of view. Many sentences are detached from the rest of the speech and the verbs should be reviewed (you are describing associations that have already occurred so the tense used should be the past).

Author Response

First, the authors sincerely thank the reviewers for agreeing to read and give us valuable contributions.

Here we would like to reply to the comments of the reviewer as follows:

  1. The purpose and innovation brought by this study should be better explained

Response 1: Dear reviewer, our research has some important new points that we would like to be published as follows:

  • The first study was conducted in Can Tho city, Vietnam, with a new biomarker in hypertensive patients (at the time we studied, this was a new biomarker in Vietnam and very few studies). After a long time fighting the COVID-19 pandemic with many difficulties, we just started writing and decided to publish.
  • At the time of the study, all patients were newly diagnosed hypertensive patients. Another study was in a group of patients diagnosed with hypertension.
  1.  It is not clear how many enrolled individuals were excluded by the criteria

Response 2: We do not show this clearly in the study diagram. We want to revise it after re-checking the study data registration table: 160 patients were included in the study and excluded: 30 patients with diabetes mellitus on sulfonylurea therapy and 15 patients with stroke at the time of diagnosis of hypertension, and 10 patients with chronic kidney disease.

  1. The study of the characteristics of the sample is not sufficient, for example, there is no information about the medications taken by the participants. This is a critical point, considering the association with diabetes and taking into account that the association between the use of metformin and elevated homocysteine levels has been widely demonstrated in the literature

Response 3: We have checked and supplemented the data from the existing records on the issue of therapeutic drugs. Because our patients were first diagnosed with hypertension, the number of cardiovascular and endocrine drugs kernel used less (table 1). At the time of the study, we analyzed further data on the association between metformin and hyperhomocysteinemia (table 2).

  1. There is no complete blood chemistry data. The levels of B vitamins, especially vitamin B12 and folate, could be decisive in explaining homocysteine levels and it is not easy to make any conclusions without such data. Mean corpuscular volume and haemoglobin levels could also help identify possible confounders

Response 4: We apologize for not having data on blood counts and substances such as vitamin B12 and folate. We want to add this point to the limitation of the study.

  1. There is no statistical evaluation that provides a model that identifies associations studied by covariates and adjusted for confounders (many of which I do not believe are in the authors' possession, as discussed above)

Response 5: After receiving comments, we recognized this limitation and requested permission to be included in the limitations and suggestions for future research.

  1. It is not possible to evaluate an association with hypertension, as all the participants were admitted for this characteristic. The presence of healthy controls would have been more helpful for this purpose. The only association that the authors can make in this regard is on the severity of hypertension, even if the population sample is probably underpowered for this purpose

Response 6: We would like to correct that this is a hypertensive stage. It was our fault (as suggested by the reviewer and another reviewer)

  1. Figure 2 is poor. It might be better to include histograms describing the frequency of hyperhomocysteinemia in various subgroups (by age, gender, etc.)

Response 7: We have edited it as a suggestion.

  1. The discussion is too verbose and focused towards mechanistic concepts which are not part of the scope of the work. Lines 215-216 are out of context.

Response 8: Sincerely sorry; we made an error when we copied the structure of MDPI from another article of ours that is currently being peer-reviewed at MDPI. We want to correct it as "Some factors associated with hyperhomocysteinemia in patients with newly diagnosed primary hypertension." And we will trim the discussion according to the feedback from the reviewer.

  1. Some passages should be revised from a linguistic point of view. Many sentences are detached from the rest of the speech and the verbs should be reviewed (you are describing associations that have already occurred so the tense used should be the past).

Response 9: We will revise it again.

Finally, on behalf of the research team, we would like to thank the reviewer for reading carefully and giving us important contributions to make our article more complete.

Round 2

Reviewer 1 Report

All Hcy should be tHcy (total homocysteine) throughout the manuscript, not in part.

Abstract: "Elevated levels of blood total homocysteine" (not homocysteine !!) is one of the risk factors for "cardiovascular diseases". The authors should mention as "Research in the subpopulation of Vietnamese people shows that..." since we know that Vietnam is a highly multiethnic community. Also in the Conclusion (line295-297).

Do not place Tables across pages!

Line 219: 1.1–14.52 (not 14, 52)

No need for "Kg/m2" for BMI. HbA1c rather than HdA1C.

The significant digits differ depending on parameters (no rules!). For example, 5.143, 12.455, 0.25 in Table 3; 0.41, 4.00, 2.693 in Table 2.

Author Response

First, the authors sincerely thank the reviewers for agreeing to read and give us valuable contributions.

Here we would like to reply to the comments of the reviewer as follows:

  1. All Hcy should be tHcy (total homocysteine) throughout the manuscript, not in part.

Response 1: We have checked and revised the manuscript.

  1. Abstract: "Elevated levels of blood total homocysteine" (not homocysteine !!) is one of the risk factors for "cardiovascular diseases". The authors should mention as "Research in the subpopulation of Vietnamese people shows that..." since we know that Vietnam is a highly multiethnic community. Also in the Conclusion (line295-297).

Response 2: We rewrote it as a reviewer's suggestion.

  1. Do not place Tables across pages!

Response 3: We have checked and revised the manuscript.

  1. Line 219: 1.1–14.52 (not 14, 52)

Response 4: We rewrote it again

  1. No need for "Kg/m2" for BMI. HbA1c rather than HdA1C.

Response 5: We rewrote it again

  1. The significant digits differ depending on parameters (no rules!). For example, 5.143, 12.455, 0.25 in Table 3; 0.41, 4.00, 2.693 in Table 2.

Response 6: We have omitted the discussion of OR to make the article clearer.

Finally, on behalf of the research team, we would like to thank the reviewer for reading carefully and giving us important contributions to make our article more complete.

Reviewer 2 Report

I appreciated the effort of the authors in editing their manuscript.

Many aspects have been clarified and implemented.

However, I recommend including the power analysis and completing the study diagram to include the total number of individuals enrolled and those excluded with reasons.

Taking into account factors related to homocysteine levels (such as the use of metformin) and in the absence of a statistical model that takes into account the various confounding factors and covariables involved, the authors could show the correlations of Table 4 also in subgroups for metformin use (users and not-users)

Author Response

First, the authors sincerely thank the reviewers for agreeing to read and give us valuable contributions.

Here we would like to reply to the comments of the reviewer as follows:

  1. However, I recommend including the power analysis and completing the study diagram to include the total number of individuals enrolled and those excluded with reasons.

Response 1: We were honored to receive a quick response from the reviewer because the sampling time was long, we stopped monitoring. We did not have more data to strengthen the statistical analysis (limitation of the study). We have redrawn the study diagram per the reviewer's feedback (we have recovered the electronic medical records and the patient's sample collection information).

  1.  Taking into account factors related to homocysteine levels (such as the use of metformin) and in the absence of a statistical model that takes into account the various confounding factors and covariables involved, the authors could show the correlations of Table 4 also in subgroups for metformin use (users and not-users)

Response 2: We analyzed spearman's test for metformin therapy as the suggestion's reviewer.

Finally, on behalf of the research team, we would like to thank the reviewer for reading carefully and giving us important contributions to make our article more complete.
